# Velocity and density characteristics of subducted oceanic crust and the origin of lower-mantle heterogeneities

Wenzhong Wang[1]*, Yinhan Xu[1], Daoyuan Sun [1,2], Sidao Ni[3], Renata Wentzcovitch[4,5,6] & Zhongqing Wu[1,2]*

Seismic heterogeneities detected in the lower mantle were proposed to be related to subducted oceanic crust. However, the velocity and density of subducted oceanic crust at lower-mantle conditions remain unknown. Here, we report ab initio results for the elastic properties of calcium ferrite-type phases and determine the velocities and density of oceanic crust along different mantle geotherms. We find that the subducted oceanic crust shows a large negative shear velocity anomaly at the phase boundary between stishovite and $CaCl_2$-type silica, which is highly consistent with the feature of mid-mantle scatterers. After this phase transition in silica, subducted oceanic crust will be visible as high-velocity heterogeneities as imaged by seismic tomography. This study suggests that the presence of subducted oceanic crust could provide good explanations for some lower-mantle seismic heterogeneities with different length scales except large low shear velocity provinces (LLSVPs).

[1] Laboratory of Seismology and Physics of Earth's Interior, School of Earth and Space Sciences, University of Science and Technology of China, Hefei, China. [2] CAS Center for Excellence in Comparative Planetology, Hefei, China. [3] State Key Laboratory of Geodesy and Earth's Dynamics, Institute of Geodesy and Geophysics, Chinese Academy of Sciences, 430077 Wuhan, China. [4] Department of Applied Physics and Applied Mathematics, Columbia University, New York, NY 10027, USA. [5] Department of Earth and Environmental Sciences, Columbia University, New York, NY 10027, USA. [6] Lamont–Doherty Earth Observatory, Columbia University, Palisades, NY 10964, USA. *email: wz30304@mail.ustc.edu.cn; wuzq10@ustc.edu.cn

The lower mantle is the largest continuous region within Earth, occupying ~55% of the volume and ~52% of the mass of the Earth, and plays a dominant role in the thermochemical and geodynamic evolution of the planet[1]. It was previously regarded as homogeneous except for the large low shear velocity provinces (LLSVPs)[2,3] and velocity anomalies near the core-mantle boundary (CMB)[4–7]. However, with the advancements in seismology, numerous heterogeneities with different length scales have been detected in the lower mantle. For instance, seismic tomography models[8–13] revealed the presence of large-scale seismic velocity anomalies in the entire lower mantle, including low-velocity columns beneath many prominent hotspots and positive velocity anomalies near subduction zones, which are regarded as hot plumes and cold slabs. Anisotropic tomography[14] further suggested that there could be complex interactions between plumes and slabs in the mid mantle. Using different types of scattering, seismological studies have also found small-scale heterogeneities with thicknesses of several or tens of kilometres and a velocity perturbation of 0.1–1% throughout the mantle[15,16]. In particular, strong scatterers with shear velocity up to ~ 12% lower than the ambient mantle are detected within a depth ranging from ~1400–1700 km[17–20] in some areas, such as regions beneath Mariana and Peru. The predominant depth of the strong scatterers significantly varies at different locations[19].

One of the unique features of the Earth is its active plate tectonics driven by vigorous mantle convection. Tomography studies imaged that some subducted slabs seem stagnant at the mantle transition zone or mid lower mantle[21], while some could reach the lowermost part of mantle[22–24]. This finding is based on the consensus that the relatively cold slabs show significantly high seismic velocities compared to the surrounding mantle, though the velocity properties of subducted materials at high P–T conditions have not yet been well investigated. Oceanic crust, which is the upper layer of the subducted oceanic lithosphere, has a quite distinctive chemical composition from the pyrolite model and is likely a major source for compositional heterogeneities in the lower mantle. Previous studies[16,19,20] have attributed the detected strong small-scale heterogeneities in the mid mantle to the phase transition from stishovite to the $CaCl_2$-type silica[25–27], which can cause a low velocity anomaly for the oceanic crust at the phase boundary[28]. However, we note that the estimated anomalies of the mid-mantle scatterers appear significantly larger than those expected for the oceanic crust[28], probably because the elastic properties of relevant materials were calculated at static conditions and hence the thermal effect cannot be taken into account. Moreover, the accumulation of subducted oceanic crust at the CMB was also speculated to play an important role in the formation of LLSVPs[29,30], around which the presence of seismic scatterers was also reported[18,31]. Consequently, the elastic and velocity properties of subducted oceanic crust at the lower-mantle conditions are crucial for interpreting the origins and evolutions of these seismic heterogeneities, modelling of small-scale mantle scattering, and evaluating the interaction between subduction and the lower mantle.

Oceanic crust is mainly composed of Mid-Ocean Ridge Basalt (MORB), which is more silicic than pyrolite. Previous experiments[32] demonstrated that the natural MORB assemblage at the P–T conditions of lower mantle consists of $SiO_2$ silica (stishovite and $CaCl_2$-type silica), calcium perovskite (CaPv), bridgmanite (Bdg), and two types of aluminum-rich phases: the new Al-rich phase (NAL) and calcium ferrite-type (CF-type). NAL and CF-type phases could coexist up to ~50 GPa, beyond which the NAL phase disappeared and only CF-type phase was identified. Further experiments[33] evaluating the phase relations of the $NaAlSiO_4$–$MgAl_2O_4$ system indicated that CF-type phase is the high-pressure polymorph of NAL phase. Combining chemical compositions and mineral volume proportions present in natural MORB[32], we find that MORB consists of approximately 39% Fe- and Al-bearing bridgmanite ($Mg_{0.58}Fe_{0.16}Al_{0.26}Si_{0.74}Al_{0.26}O_3$), 30% Ca-perovskite ($CaSiO_3$), 16% $SiO_2$, and 15% Fe-bearing CF-type phase ($Na_{0.4}Mg_{0.48}Fe_{0.12}Al_{1.6}Si_{0.4}O_4$). Therefore, the elastic properties of these minerals, which are sensitive to the incorporation of substitutional solutes, are of great importance for the determination of velocity and density of MORB. Previous ab initio calculations within the local density approximation (LDA) have obtained the reliable and accurate elasticity of Fe-free and Fe-bearing bridgmanite ($MgSiO_3$ and $Mg_{0.875}Fe_{0.125}SiO_3$)[34], corundum ($Al_2O_3$)[35], Ca-perovskite[36], and stishovite and $CaCl_2$-type silica[37] at high P–T conditions. However, the elastic properties of CF-type phase have not been investigated under lower-mantle conditions.

Here we obtain the elastic properties of CF-type phase at high pressure and temperature using ab initio calculations. Combining our results with previous studies, we determine the velocities and density of subducted oceanic crust under lower-mantle conditions. Our results show that the velocity anomalies produced by subducted oceanic crust strongly depend on depth and its presence can explain some seismic heterogeneities in the lower mantle.

## Results

**Equation of state of CF-type phases.** We calculated the elastic properties of two end-members of CF-type phases ($NaAlSiO_4$ and $MgAl_2O_4$) and considered the iron incorporation ($Mg_{0.75}Fe_{0.25}Al_2O_4$) using the same methodology as our previous studies (see methods). As shown in Fig. 1, the predicted pressure-dependent volumes agree well with available experimental measurements at 300 K[38–44], and the largest discrepancy is <1%, except some data from Dubrovinsky et al. (2002)[45], which deviate from other experimental data and our LDA calculations by up to ~2.5%. We note that the experimentally measured density of $Na_{0.4}Mg_{0.6}Al_{1.6}Si_{0.4}O_4$ CF-type phase is also consistent with our results up to ~80 GPa, above which experimental data slightly deviate from our results (Supplementary Fig. 1). These comparisons clearly demonstrate the high reliability of our results.

**Elastic and velocity properties of CF-type phases.** The calculated bulk moduli ($K_S$), shear moduli ($G$), compressional velocities ($V_P$), and shear wave velocities ($V_S$) of CF-type minerals at various pressures and temperatures are shown in Fig. 2. The pressure- and temperature-dependent elastic tensors are presented in Supplementary Fig. 2. Our results suggest that different CF-type phases have similar pressure and temperature dependences for $K_S$, $G$, $V_P$, and $V_S$ (Fig. 2), consistent with Zhao et al. (2018)[46]. The temperature dependences of these properties are almost linear but are significantly weakened at high pressure (Fig. 2). For instance, the first temperature derivatives at 30 GPa ($\partial K_S/\partial T = -1.72 \times 10^{-2}$, $\partial G/\partial T = -1.48 \times 10^{-2}$ GPa $K^{-1}$, $\partial V_P/\partial T = -2.63 \times 10^{-4}$, and $\partial V_S/\partial T = -1.98 \times 10^{-4}$ km s$^{-1}$ $K^{-1}$) are markedly lower than those at 100 GPa ($\partial K_S/\partial T = -1.17 \times 10^{-2}$, $\partial G/\partial T = -0.97 \times 10^{-2}$ GPa $K^{-1}$, $\partial V_P/\partial T = -1.00 \times 10^{-4}$, and $\partial V_S/\partial T = -0.87 \times 10^{-4}$ km s$^{-1}$ $K^{-1}$). In contrast, noticeable nonlinear dependences on pressure are observed for elastic moduli and wave velocities, especially $V_P$ and $V_S$ (Fig. 2 and Supplementary Table 1). At 2000 K, the first pressure derivatives, $\partial K_S/\partial P$, $\partial G/\partial P$, $\partial V_P/\partial P$, and $\partial V_S/\partial P$, decrease from 3.82, 1.70, 43.47 km s$^{-1}$ MPa$^{-1}$, 20.74 km s$^{-1}$ MPa$^{-1}$ at 30 GPa to 3.41, 1.17, 23.42 km s$^{-1}$ MPa$^{-1}$, and 8.96 km s$^{-1}$ MPa$^{-1}$ at 100 GPa, respectively.

The chemical composition affects the elastic moduli and wave velocities of CF-type phases. Compared to $MgAl_2O_4$, $NaAlSiO_4$ has a slightly smaller Ks but larger G. The contrasts in Ks and G

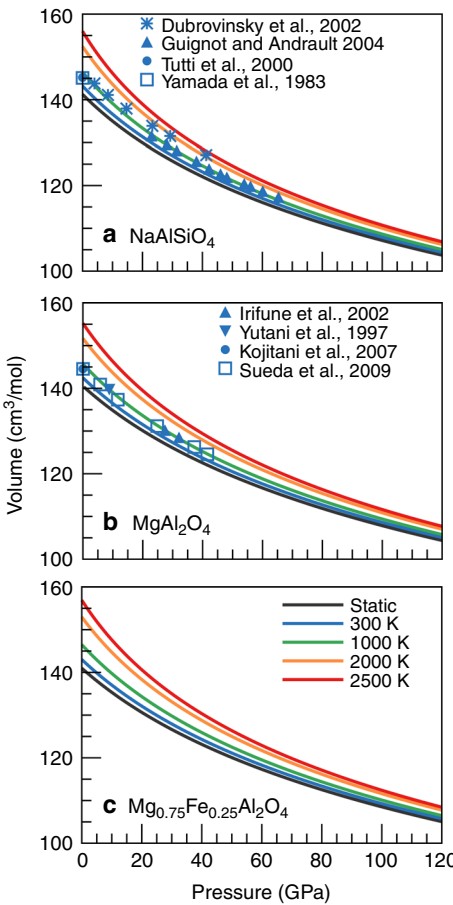

**Fig. 1 Compression curves of CF-type phases. a** $NaAlSiO_4$, **b** $MgAl_2O_4$, and **c** $Mg_{0.75}Fe_{0.25}Al_2O_4$. Colorful lines represent ab initio results at variable temperatures and points are experimental data in previous studies[38-45].

among these end-members range from ~−4% to ~0% and from ~4% to ~1% respectively, when pressure increases from 30 to 100 GPa (Fig. 2 and Supplementary Fig. 3). Therefore, the velocities of these CF-type phases differ by 0.5–2.5% for $V_S$ and by less than 1% for $V_P$. Notably, these differences are evidently diminished at high pressure (Supplementary Fig. 3). In addition, the incorporation of 25 mol% of iron into $MgAl_2O_4$ does not significantly affect $K_S$ but obviously decreases G, e.g., by 5.5% at 60 GPa and 2000 K, which causes a reduction of 3.3% in $V_P$ and of 5% in $V_S$. These reductions are amplified by temperature but lessened by pressure. The wave velocities and densities of CF-type phases along the normal mantle geotherm[47] are compared with those of other lower-mantle minerals in Fig. 3. $NaAlSiO_4$ and $MgAl_2O_4$ have similar $V_P$ and $V_S$ that are ~2.8–6.0% and 2.6–5.8% lower than those of bridgmanite, respectively, although their densities are almost identical to that of bridgmanite. Instead, $Mg_{0.75}Fe_{0.25}Al_2O_4$ has a relatively higher density and much lower $V_P$ and $V_S$. The maximum differences in $V_P$, $V_S$, and density between $Mg_{0.75}Fe_{0.25}Al_2O_4$ and bridgmanite are up to −8%, −10%, and 4%, respectively. On the basis of our results for these three end-members, we can obtain the elasticity and velocity of CF-type phase with different compositions, such as $Na_{0.4}Mg_{0.6}Al_{1.6}Si_{0.4}O_4$, whose G is consistent with experimental data[48] at 300 K.

## Discussion

MORB with a distinctive chemical composition from the normal mantle is likely a major source for small-scale scatterers with the thickness of several kilometres in the lower mantle. Seismic modelling of whole-mantle small-scale scattering suggested a length scale of ~8 km and a wide range of velocity perturbations varying from ~0.1% to ~1% under the assumption that heterogeneities were randomly and evenly distributed throughout the mantle[15,16]. Combining our results with previous data[34–37], we estimated the velocities and density of the natural MORB assemblage[32] at the lower-mantle conditions. We find that MORB has higher velocities than the ambient mantle through most of the lower mantle but lower velocities than the ambient mantle at the mid mantle depths (see Fig. 4). Particularly, the velocity contrasts between MORB and the normal mantle are extremely sensitive to the depth at the mid mantle. Thus, velocity heterogeneities caused by the presence of MORB are noticeably depth-dependent and unevenly distributed throughout the mantle even if MORB distributes evenly throughout the mantle. This finding is more complicated than the single layer heterogeneity model adopted by Bentham et al. (2017)[16]. In addition, the density/velocity fluctuation scaling factors of MORB also strongly depend on depth. Because MORB is denser than ambient mantle, scaling factors even become negative at the mid mantle where the velocity perturbations are negative. Thus, the simple density/velocity fluctuation scaling factors that have been widely used to model the small-scale scattering throughout deep Earth[15] are not valid for MORB. How the depth-dependent velocity heterogeneities and density/velocity fluctuation scaling factors affect the seismic scattering modelling is worth investigating.

MORB has distinctly slower wave velocities by up to ~ −7% for $V_S$ and ~ −1.8% $V_P$ at ~60 GPa (Fig. 4d, e), where stishovite transforms to $CaCl_2$-type silica[25]. Such large negative velocity anomalies within MORB, which are mainly caused by softening of the shear modulus of stishovite at the phase boundary[37], provide good explanations for the observed seismic scatterers or small-scale heterogeneities with quite low shear velocities[19,20] in the mid mantle (Fig. 5). Remarkably, the observed $V_S$ anomalies of the mid mantle scatterers[19,20] are consistent with those from our mineralogical predictions, despite the large uncertainties in seismological estimates. Tsuchiya (2011)[28] predicted negative $V_P$ and $V_S$ contrasts between MORB and pyrolite at the phase boundary of silica; however, the depth for such negative velocity perturbations is significantly shallower than our results, mainly because the temperature effect has not been taken into account[28]. We obtained similar results (negative velocity anomalies and their depths) in Tsuchiya (2011)[28] when only elastic data at static conditions were used. The magnitude of the maximum velocity anomaly for MORB is not significantly sensitive to the temperature variation (Fig. 4), but the depth where the maximum velocity anomaly occurs is mainly controlled by the phase boundary between stishovite and $CaCl_2$-type silica, which strongly depends on temperature and alumina and water contents bearing in silica[25]. The incorporation of alumina plus hydrogen into silica can strikingly decrease the transition pressure, while increasing temperature significantly elevates it due to the positive Clapeyron slope[25]. Therefore, the variations of bright depth for scattering[17–19], where the strong seismic scatterers were observed in the mid-lower mantle, may reflect the differences in temperature and $Al_2O_3$ and $H_2O$ contents of silica in the oceanic crust.

After the phase transition of silica, MORB along the normal geotherm has relatively higher wave velocities than the ambient mantle (Fig. 3). The $V_P$ and $V_S$ perturbations are up to ~ +1.8% and +1.2%, respectively. If assuming a temperature anomaly of −500 K existing in MORB, the positive $V_P$ and $V_S$ anomalies increase to +2.5% and +2.0% (Fig. 4d, e), respectively. Stixrude and Lithgow-Bertelloni (2012)[49] also found that MORB has a faster $V_S$ than the pyrolitic composition along the normal mantle

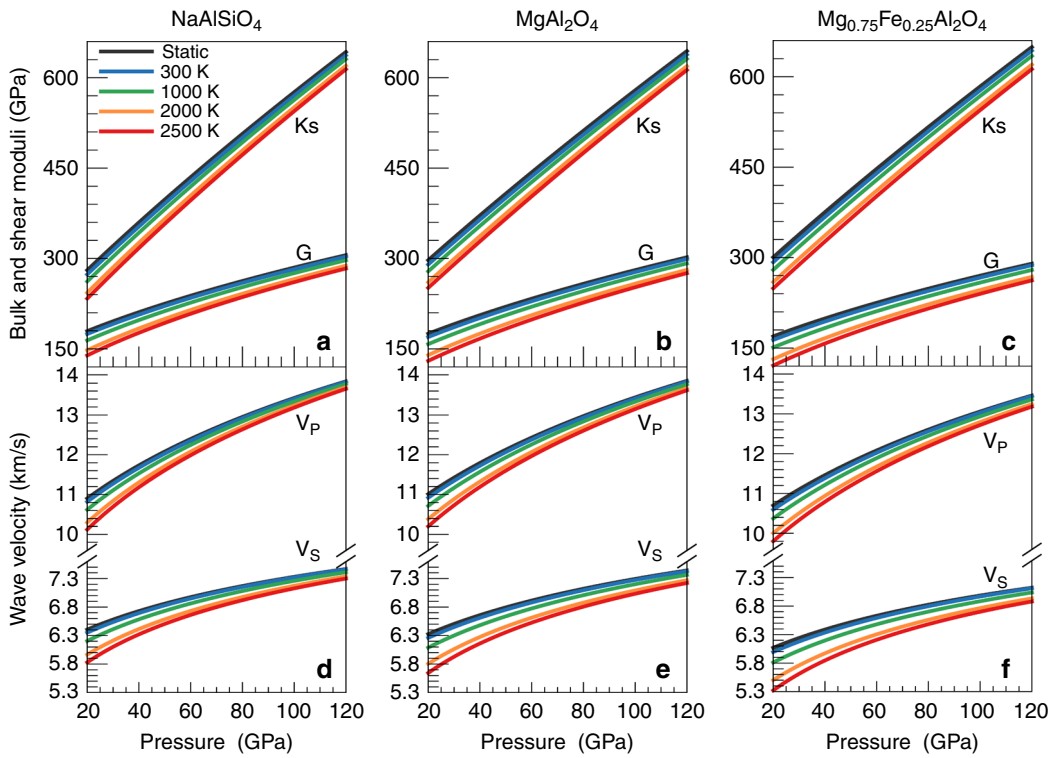

**Fig. 2 Elastic moduli and wave velocities of CF-type phases. a–c** bulk and shear moduli ($K_S$ and G), **d–f** compressional and shear wave velocities ($V_P$ and $V_S$). Elastic moduli and wave velocities for **a, d** $NaAlSiO_4$, **b, e** $MgAl_2O_4$, **c, f** $Mg_{0.75}Fe_{0.25}Al_2O_4$.

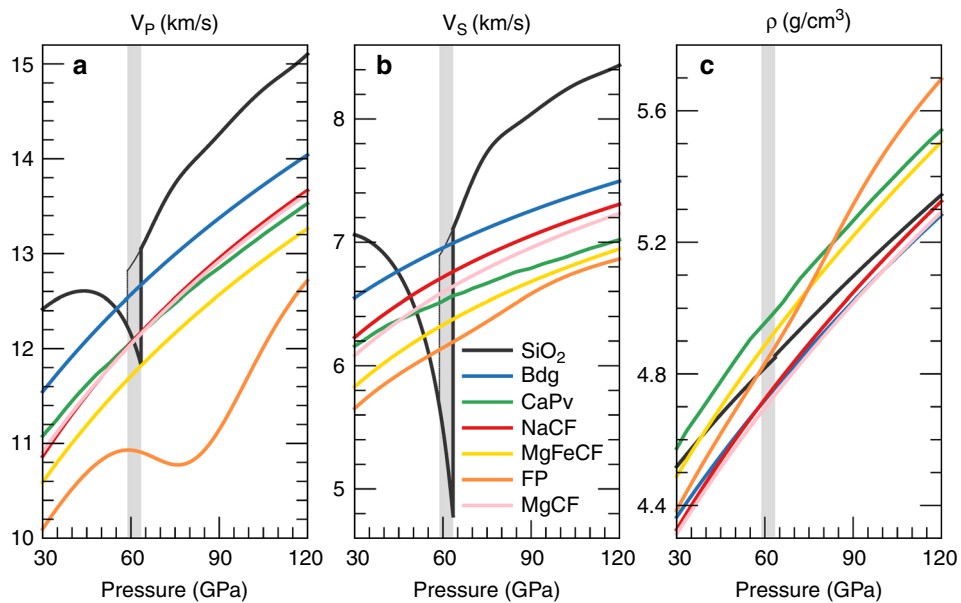

**Fig. 3 Comparisons of velocities and density between CF-type phases and other lower-mantle minerals along the normal mantle geotherm. a** compressional wave velocities ($V_P$), **b** shear wave velocity ($V_S$), and **c** density (ρ). The normal mantle geotherm is derived from Brown and Shankland (1981)[47]. Data sources: NaCF, $NaAlSiO_4$ CF-type phase, this study; MgCF, $MgAl_2O_4$ CF-type phase, this study; MgFeCF, $Mg_{0.75}Fe_{0.25}Al_2O_4$ CF-type phase, this study; $SiO_2$, stishovite and the $CaCl_2$-type silica, Yang and Wu (2014)[37]; Bdg, $Mg_{0.92}Fe_{0.08}SiO_3$ bridgmanite, Shukla et al. (2015)[34]; CaPv, Ca-perovskite, Kawai and Tsuchiya (2015)[36]; FP, $Mg_{0.82}Fe_{0.18}O$ ferropericlase, Wu et al. (2013)[64]. Grey areas represent the calculated phase boundary between stishovite and the $CaCl_2$-type silica[37].

geotherm at the depth range of 1500–2500 km, and the $V_S$ contrast is comparable to our results; however, they did not find the shear softening of MORB at the mid mantle because they ignored the phase transition from stishovite to $CaCl_2$-type silica. Seismic tomography[8–12] indicates that the positive shear velocity anomalies in the mid-lower mantle beneath some local regions, which are generally regarded as the presence of subducted slab, can be larger than +1.5% and even to +2%. Such a large positive

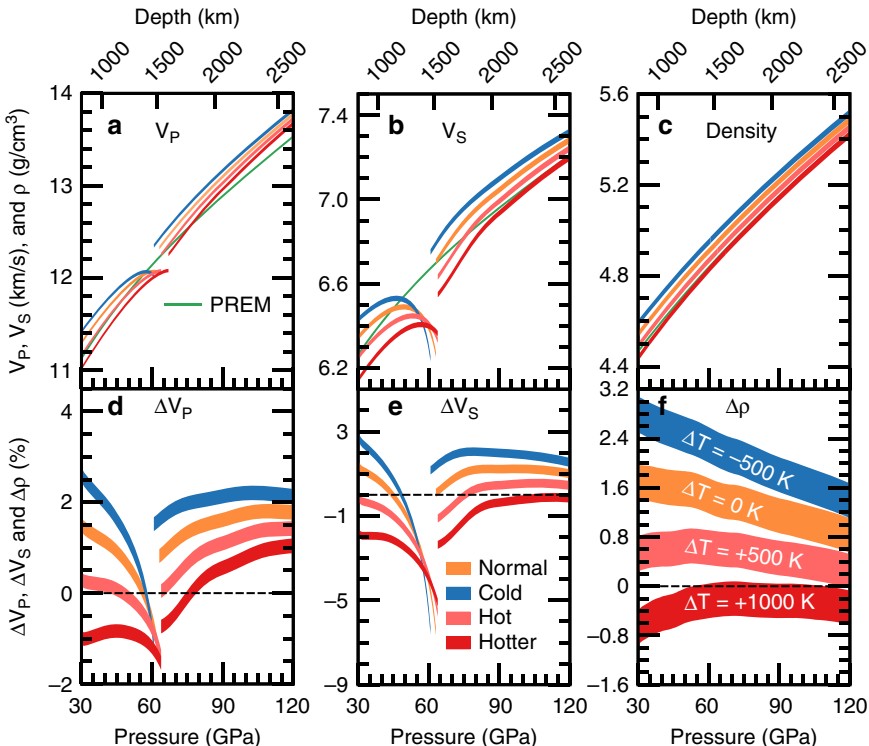

**Fig. 4 Velocities and density characteristics of subducted oceanic crust. a–c** Wave velocities ($V_P$ and $V_S$) and densities ($\rho$) of MORB along different mantle geotherms. The normal mantle geotherm is from Brown and Shankland (1981)[47]. Red, pink, orange, and blue lines represent $V_P$, $V_S$, and $\rho$ of MORB along the mantle geotherms with variable temperature anomalies relative to the normal mantle geotherm. Temperature anomalies for red, pink, orange, and blue lines are + 1000 K, + 500 K, 0 K, and 500 K, respectively. Green lines are PREM values[69]. MORB[32]: 39% Fe- and Al-bearing bridgmanite ($Mg_{0.58}Fe_{0.16}Al_{0.26}Si_{0.74}Al_{0.26}O_3$), 30% Ca-perovskite ($CaSiO_3$), 16% $SiO_2$, and 15% Fe-bearing CF-type phase ($Na_{0.4}Mg_{0.48}Fe_{0.12}Al_{1.6}Si_{0.4}O_4$). (d) (e) (f) the $V_P$, $V_S$, and $\rho$ contrasts between MORB and PREM. $\Delta M = 2(M_{MORB}-M_{PREM})/(M_{MORB} + M_{PREM})$, $M = V_P$, $V_S$, and $\rho$. $M_{PREM}$, green lines in **a**, **b**, **c**; $M_{MORB}$, red, orange, or blue lines in **a**, **b**, **c**. The linewidth represents uncertainties caused by the errors for elastic properties (<0.8%) and the variations in the concentration of dilute substitutional solutes (±1 mol%).

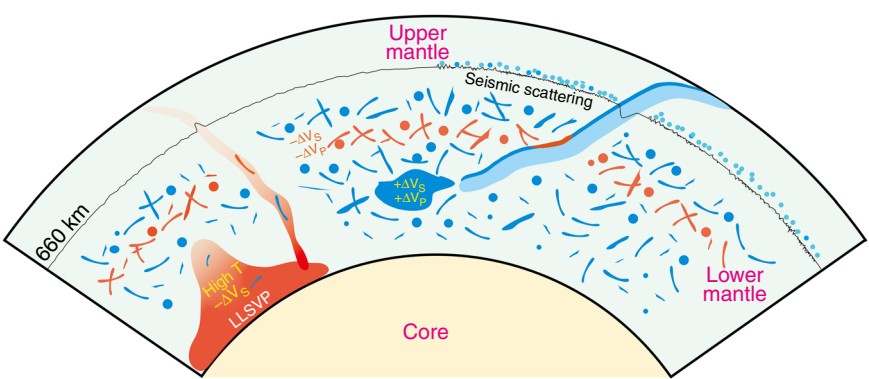

**Fig. 5 Schematic diagram for subducted oceanic crust in the lower mantle.** The velocity heterogeneities caused by subducted oceanic crust are noticeably depth-dependent: it produces large negative velocity anomalies at the mid mantle but high velocity heterogeneities at the lower part of mantle. The presence of subducted oceanic crust could provide explanations for seismic scatters and high velocity heterogeneities (~2%) in the lower mantle imaged by seismic tomography, but LLSVPs likely do not originate from subducted oceanic crust. The 660-km discontinuity, which defines the top of the lower mantle, was also found to show the small-scale topographic variations[71].

velocity anomaly cannot be simply caused by temperature variations alone and must have also a compositional origin. For example, a temperature reduction of 500 K can only increase the $V_P$ and $V_S$ of a pyrolitic composition by <1% (Supplementary Fig. 4). Our results suggest that MORB is an important candidate for these compositional heterogeneities. MORB could perhaps accumulate in the mid lower mantle, since there may be some barriers for subduction such as an increase in viscosity induced by

the spin transition of iron[50] or the ancient mantle high-viscosity structures[51]. Furthermore, our results also confirm that MORB is denser than the surrounding mantle at lower mantle pressures (Fig. 4c). The excess density perturbation decreases with depth and is ~ +1.4% on average (Fig. 4f) when MORB has the same temperature to the ambient mantle, consistent with the previous estimations[32,52]. The negative buoyancy of the oceanic crust plays a key role in its descent to the CMB, which is also revealed by

seismic tomography[8–10]. However, recent geodynamical simulations argued that not all of the subducted oceanic crust would accumulate at the CMB because the negative buoyancy provided by this thin domain is not quite sufficient to overcome viscous forces[53]. Therefore, from a geodynamics perspective, the partial accumulation of MORB likely occurs at lower-mantle depths far above the CMB and generates seismologically well-known high velocity heterogeneities (Fig. 5).

In addition, the accumulation of subducted oceanic crust at the CMB was also speculated as a possible origin for LLSVPs[29], the thermochemical heterogeneities characterized by slow shear wave velocities[2,3]. Coincidentally, seismological studies detected small-scale scatterers above the Pacific LLSVP and near the edge of the African LLSVP[18,31]. However, our results demonstrate that the $V_P$ and $V_S$ of MORB are at least 0.6% higher than those of the ambient mantle even when the temperature anomaly within MORB is +500 K (Fig. 4d, e), and negative velocity anomalies can be produced only when MORB is at least +1000 K hotter than the ambient mantle. This is obviously in contradiction with the large negative $V_S$ anomalies (up to −3%) within LLSVPs[2,3], thus indicating that the distinct composition of LLSVPs unlikely originates from the subducted oceanic crust. The spatial distributions of seismic scatterers around LLSVPs might coincidentally result from the mantle convection. Nonetheless, it is still unknown whether the fragments of subducted oceanic crust remain in LLSVPs, and further detections of inside seismic scatterers probably will help to clarify this problem.

A recent work conducted by Thomson et al. (2019)[54] suggested that subducted oceanic crust would be visible as low-seismic-velocity anomalies throughout the lower mantle when data are extrapolated to the lower-mantle conditions. The discrepancy between our results and Thomson et al. (2019)[54] resulted from the usage of different elastic and velocity data for Ca-perovskite. The calculated data used in this study are from previous ab initio molecular dynamic simulations[36], while the Ca-perovskite data adopted in Thomson et al. (2019)[54] were extrapolated from low pressure to the deep mantle conditions. Since velocities measured for Ca-perovskite[54,55] are considerably lower than computational predictions at the conditions of the uppermost lower mantle, the extrapolated data would be expectedly lower than theoretical calculations[36] under deep mantle conditions. It is still unknown what results in the discrepancies in sound velocities of Ca-perovskite between theoretical and experimental studies, and future research is needed to solve this problem. However, the uncertainties from extrapolation cannot be ignored because experimental measurements, especially high-temperature data, which also shows significant discrepancies[54,55].

The presence of subducted oceanic crust in the lower mantle can provide good explanations for some detected velocity heterogeneities with different length scales, indicating the cycling of crustal materials into the deep mantle (Fig. 5). The velocity and density characteristics of subducted oceanic crust support that it could not only produce a number of remnant fragments with several kilometer thicknesses that are detected by seismic scattering[19] in the lower mantle but may also partially accumulate in the mid lower mantle or at the CMB to form the mesoscale chemical heterogeneities with positive velocity anomalies[8,9]. How the subducted oceanic crust produces the seismic heterogeneities with different length scales can be further evaluated by using the geodynamic modelling for the interaction between the subducted slab and the lower mantle. Furthermore, geodynamic simulations[56] suggested that the subducted oceanic crust would also be entrained into mantle plumes, inducing the geochemical complexity of hotspot lavas. In that case, the basaltic fragments involved in the mantle plume would probably be detected by seismic scattering, which could independently validate the relationship between the geochemical heterogeneity and oceanic crust. The current scenario of subducted oceanic crust in the lower mantle provides important clues about the interaction between the subducted slab and the lower mantle and the thermochemical evolution of the lower mantle.

## Methods

**First-principles calculations**. Ab inito calculations were performed using Quantum Espresso package[57] based on the density functional theory (DFT), plane waves, and pseudopotentials. The local density approximation (LDA) was adopted as the exchange correlation functional. Pseudopotentials for magnesium, silicon, aluminum, and oxygen used in this study are well described in previous studies[37,58,59]. The pseudopotentials for sodium and iron were generated by Vanderbilt method[60] with a valence configuration of $2s^22p^63s^1$ for Na, and $3s^23p^63d^{6.5}4s^14p^0$ for Fe. The energy cutoff for plane waves was 70 Ry and the Brillouin zone for the electronic state summation was sampled on a $2 \times 2 \times 10$ mesh for CF-type phase ($NaAlSiO_4$, $MgAl_2O_4$, and $Mg_{0.75}Fe_{0.25}Al_2O_4$). To sufficiently describe the large on-site Coulomb interactions among the Fe 3d electrons in the Fe-bearing CF-type phase ($Mg_{0.75}Fe_{0.25}Al_2O_4$), we used the LDA + U method, introducing a Hubbard U correction to the LDA. The U value for ferrous Fe in CF-type phase is 2.7 eV, which was non-empirically determined using the linear response method[61]. Structures of CF-type phase were well optimized at variable pressures using the variable cell-shape damped molecular dynamics approach[62]. Vibrational density of states (VDoS) at different equilibrium volumes were calculated using the finite displacement method. The elastic tensors at static conditions were derived from the linear relationship between stress and strain. The strain magnitude applied to relaxed structures was 1%.

**Elasticity of CF-type phases at high pressure and temperature**. The usual method used to calculate the elasticity at high temperature and pressure usually needs lots of vibrational density of states of material under different volumes and different strains[63], which requires huge computational effort and hampers the accurate numerical evaluations of elastic properties at high P–T conditions. Wu and Wentzcovitch (2011)[63] developed a semi-analytical approach without requiring the vibrational density of states under strain by analyzing the relation between volume dependence of and strain dependence of the vibrational frequencies, which reduces the computational workload by one order of magnitude compared to the usual method without loss of accuracy. This method has been also successfully applied to bridgmanite[34], ferropericlase[64], stishovite and CaCl2-type silica[37], and corundum[35]. In this work, based on the elastic tensors at static conditions and VDoS at variable equilibrium volumes, we also calculated elastic properties of CF-type phases at high pressure and temperature using this semi-analytical approach. The adiabatic bulk modulus $K_S$ and shear modulus G were obtained by computing the Voigt-Reuss-Hill averages[65] from elastic tensors. Thus, compressional and shear velocities were calculated from the equations $V_P = \sqrt{(K_S + \frac{4}{3}G)/\rho}$ and $V_S = \sqrt{G/\rho}$ ($\rho$ is density).

To estimate the effect of pseudopotentials' quality on elastic properties, we also used harder pseudopotentials that includes semicore states with smaller core radii to conduct static calculations, which requires a cutoff energy of 400 Ry. We found that there are only minor differences (<0.3%) in elastic moduli and density at static conditions when different pseudopotentials were used. Previous studies[59] on elastic properties and density of bridgmanite also reported minor differences in the results obtained using different pseudopotentials. The intrinsic anharmonicity ignored by quasi-harmonic approximation (QHA) should be negligible at the lower-mantle pressure and temperature conditions and can be estimated by the difference in the results calculated from QHA and molecular dynamic (MD) simulations. By comparing the elastic moduli and density of $MgSiO_3$ bridgmanite obtained from QHA and MD within LDA[66], we also found only minor differences (<0.5%) produced by the anharmonicity at high pressures. Thus, these differences produced by different pseudopotentials and the anharmonic effect were adopted as the uncertainties of the calculated results (<0.8%).

**Elastic moduli and velocities of MORB**. On the basis of the elastic properties of bridgmanite, ferropericlase, and Ca-perovskite at the conditions of the lower mantle, previous studies[59,67,68] suggested that a pyrolitic composition can reproduce the reference velocities and densities of PREM[69]. An appropriate and likely composition for a pyrolitic lower mantle[67,68,70] is likely composed of 15% ferropericlase ($Mg_{0.82}Fe_{0.18}O$), 78% Fe-bearing bridgmanite ($Mg_{0.92}Fe_{0.08}SiO_3$), and 7% Ca-perovskite ($CaSiO_3$), which is adopted in this work. The MORB composition[32] consists of ~39% Fe- and Al-bearing bridgmanite ($Mg_{0.58}Fe_{0.16}Al_{0.26}Si_{0.74}Al_{0.26}O_3$), 30% Ca-perovskite ($CaSiO_3$), 16% $SiO_2$, and 15% Fe-bearing CF-type phase ($Na_{0.4}Mg_{0.48}Fe_{0.12}Al_{1.6}Si_4O_4$).

The elastic properties of Ca-perovskite, Fe-free and Fe-bearing bridgmanite ($MgSiO_3$ and $Mg_{0.875}Fe_{0125}SiO_3$), corundum ($Al_2O_3$), ferropericlase ($Mg_{0.82}Fe_{0.18}O$), stishovite, and the CaCl2-type silica at high P–T conditions are reported in previous theoretical studies[34–37,64]. Combining these data with our elastic data of CF-type phase ($NaAlSiO_4$, $MgAl_2O_4$, and $Mg_{0.75}Fe_{0.25}Al_2O_4$), we

calculated elastic moduli and densities of all the above phases with chemical compositions as shown in the MORB using the interpolation method. Thus, elastic moduli and densities of the MORB were calculated using:

$$\rho = \sum_i f_i \rho_i \tag{1}$$

$$M = \frac{1}{2}\left[\sum_i f_i M_i + \left(\sum_i f_i M_i^{-1}\right)^{-1}\right] \tag{2}$$

where $\rho_i$, $M_i$, and $f_i$ are the density, bulk modulus ($K_S$) or shear modulus (G), and the fraction of the $i$th mineral, respectively. Then, the compressional and shear velocities ($V_P$ and $V_S$) were derived from $V_P = \sqrt{(K_S + \frac{4}{3}G)/\rho}$ and $V_S = \sqrt{G/\rho}$. The uncertainties of calculated velocities and density of MORB were estimated based on the errors for elastic properties (<0.8%) and the concentration of dilute substitutional solutes (±1 mol%).

## Data availability

The data sets in this study are available as Supplementary Information and from the corresponding authors.

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

## Acknowledgements

This study is supported by the Strategic Priority Research Program (B) of the Chinese Academy of Sciences (XDB18000000), State Key Development Program of Basic Research of China (2014CB845905), National Natural Science Foundation of China (41721002), the Fundamental Research Funds for the Central Universities (WK2080000078), and Special Program for Applied Research on Super Computation of the NSFC-Guangdong Joint Fund (the second phase). The calculations were partly conducted at supercomputing center of USTC. The data were generated by Quantum Espresso (www.quantumespresso.org) and listed in the Supplementary Materials.

## Author contributions

Z.W. and W.W. designed this project. W.W. and Y.X. performed ab initio calculations, modelled the results, and wrote the draft. W.W., Y.X., Z.W., D.S., S.N. and R.W. contributed to the revision of the manuscript. W.W. and Y.X. contributed equally to this work.

## Competing interests

The authors declare no competing interests.
