## [Peer Review File · Nature Communications]

Reviewers' comments:

Reviewer #1 (Remarks to the Author):

Comments on "Velocity and density characteristics of subducted oceanic crust and the origin of seismic heterogeneity in the lower mantle" by Wang et al.

This paper presents results of *ab initio* calculations of density and elastic properties of basalt under the lower mantle pressure and temperature conditions. The topic is quite important for understanding the dynamics of deep earth. Since there are only a few studies that numerically evaluate seismic anomalies of basaltic rock, and since it is still difficult to measure the wave speeds in the laboratory, I think this paper can be accepted if it is improved in terms of the points I raise below.

1. The uncertainty ranges of the calculated wave speeds and density need to be evaluated and presented clearly, since they are highly useful for researchers of other fields when they refer to this study.
2. Results of previous studies about the same topic, such as Tsuchiya (2011, PEPI) and Stixrude & Lithgow-Bertelloni (2012, *Annu. Rev. Earth and Planet. Sci.*), need to be assessed and compared carefully with the results of this study.
3. This point is linked with the previous comment. Main part of this study is about the computation of seismic properties of the CF type phase. It is of importance, but the predicted behavior of this mineral turns out not to have controlling effects on the properties of basalt as a whole, unlike stishovite. Therefore the authors need to clarify the most important contribution of this paper to the topic dealt with. They also need to clarify technical barriers that have hampered accurate numerical evaluations of elastic properties of this mineral, and the innovative aspects of their approach to overcome the difficulties.

Minor points:

1. (Line 59): The scatterer mentioned is not beneath Tonga but beneath northern Peru (Haugland, et al, 2017).
2. (Figure 1) : The symbols of experimental studies are rather difficult to see in Figure 1.
3. (Figures 4d and 4e): A horizontal line at $dV_p=0$ and $dV_s=0$ seems to help understand the points mentioned in the text.

Reviewer #3 (Remarks to the Author):

This is an interesting study focusing on a very timely topic: the physical properties of subducted oceanic crust and their implications regarding compositional heterogeneity in the lower mantle. There is currently a strong debate about the lower mantle, which seems to be more complex and heterogeneous than previously thought. The mineral physics calculations performed in this study are much-needed for the interpretation of geophysical observations and models. However, I have some concerns as highlighted below.

1. I am concerned that none of the figures seem to show uncertainties, so it is difficult to understand how well constrained the various properties are. The methods section should include some brief explanation of the uncertainties affecting the calculations and, more broadly, of the limitations of the methods employed. Clear statements on which properties are well/less well constrained should be included.

2. Linked to the previous point, when reading the text comparing the new calculations with experimental data (lines 107-111), the reader is left unclear about the reasons for discrepancies with the experimental data of Dubrovinsky et al., 2002 and why one should favor the measurements of Guignot and Andraut, 2004 (in practice it is really mostly these two studies that are being compared). Likewise, the reasons for discrepancies with experimental data at high pressure (Fig. S1) are not entirely clear. Again, this sort of brief discussion could be included in the Methods section along with a discussion of uncertainties and limitations, to give more confidence on the results presented and whether they really support the conclusions stated.

Other comments:

3. Lines 134-135: why is there a larger contrast in V_s than in V_p ?

4. Fig. 4: of course, it is common practice to perform comparisons with PREM, but it could be useful to also carry out comparisons with 1-D profiles from various locations (e.g., near subducted slabs) from recent seismic tomographic models (e.g., Fukao and Kobayashi, 2013 for V_p ; Auer et al., 2014 and Chang et al., 2015 for V_s).

5. In order to make the text more accessible to a broad audience, it would be useful to add here and there the depth range that the results apply to (e.g., in line 30 of the abstract, and throughout the whole manuscript). For example, in which range of mid-mantle depths MORB may get lower velocities than ambient mantle? Which lateral variations may be expected?

6. In order to make a stronger case on the importance and complexity of the lower mantle, the authors should also refer to recent studies of seismic anisotropy - e.g., in line 54 it could be added that there are not only plumes and slabs, but they can even complex interactions in the mid-mantle. For example, Chang et al., Nat. Comms., 2016 showed such example of complex interactions and it should be mentioned here.

7. In order to make the study more appealing and easier to follow to a broad audience, it would be useful to include a final cartoon figure summarising observations of velocity heterogeneity that could potentially be explained in the framework of the calculations presented in this study.

8. Overall the text needs to be improved; there are quite a few typos and the text should be sharper. For example, the sentence in lines 161-165 needs to be re-written for improved clarity, in line 99 write "ab initio CALCULATIONS", in line 141 write "with those" instead of "with these", in line 166 write "depend" instead of "depends", etc (these are just some examples, a thorough revision of the manuscript is needed).

9. The last sentence of the main manuscript is quite vague - how could the calculations performed in this study help constrain, e.g., the vigour of mantle convection? More specific arguments would make the study stronger.

References:

Auer, L., Boschi, L., Becker, T. W., Nissen-Meyer, T. and Giardini, D. (2014) Savani: A variable resolution whole-mantle model of anisotropic shear velocity variations based on multiple data sets, *J. Geophys. Res.*, 119, 3006-3034, doi:10.1002/2013JB010773.

Chang, S.-J., Ferreira, A. M. G., Ritsema, J., van Heijst, H. J. and Woodhouse, J. H. (2015) Joint inversion for global isotropic and radially anisotropic mantle structure including crustal thickness perturbations, *J. Geophys. Res.*, 120, 4278-4300.

Chang, S.-J., Ferreira, A. M. G., Faccenda, M. (2016). Upper- and mid-mantle interaction between the Samoan plume and the Tonga-Kermadec slabs. *Nature Communications*, 7 10799. doi:10.1038/ncomms10799

Fukao, Y., Obayashi, M. (2013) Subducted slabs stagnant above, penetrating through, and trapped below the 660 km discontinuity. *J. Geophys. Res.*, 118, 5920-5938, doi:10.1002/2013JB010466.

Reviewer #4 (Remarks to the Author):

The paper reports on a study of the elastic properties of calcium ferrite-type phases with substitutional solutes, to determine the seismic velocities and densities of oceanic crust. The study is based on first principles methods, within the density functional theory implementation of quantum mechanics, and in particular by including the Hubbard U to address strong correlation.

The paper contains an extensive set of calculations, and the results appear interesting, though I will not comment on their geophysical relevance. The first principles calculations have been validated on available experimental data, and so there is strong support that they are reliable. The authors are using tested methods, and so there would seem to be little doubt that these calculations are state of the art. However, very little detail of the molecular dynamics simulation is provided, and so it is difficult to make an informed judgement.

The manuscript should also be checked for editing problems, e.g.:

- line 258 "insufficiently describe"

Reviewer #1

Comment 1

This paper presents results of ab initio calculations of density and elastic properties of basalt under the lower mantle pressure and temperature conditions. The topic is quite important for understanding the dynamics of deep earth. Since there are only a few studies that numerically evaluate seismic anomalies of basaltic rock, and since it is still difficult to measure the wave speeds in the laboratory, I think this paper can be accepted if it is improved in terms of the points I raise below.

[Authors]: Thanks for the recommendation. Please see our replies below.

Comment 2

The uncertainty ranges of the calculated wave speeds and density need to be evaluated and presented clearly, since they are highly useful for researchers of other fields when they refer to this study.

[Authors]: Thanks for the suggestion. We estimated the uncertainties for the calculated elasticity and density of minerals by checking the quality of pseudopotentials used in this work and comparing the results of MgSiO₃ bridgmanite obtained from the quasi-harmonic approximation (QHA) and molecular dynamic (MD) simulations within LDA. Please refer to the contents in lines 295-308:

"To estimate the effect of pseudopotentials' quality on elastic properties, we also used harder pseudopotentials that includes semicore states with smaller core radii to conduct static calculations, which requires a cutoff energy of 400 Ry. We found that there are only minor differences (< 0.3%) in elastic moduli and density at static conditions when different pseudopotentials were used. Previous studies⁵⁹ on elastic properties and density of bridgmanite also reported minor differences in the results obtained using different pseudopotentials. The intrinsic anharmonicity ignored by QHA should be negligible at the lower-mantle pressure and temperature conditions and can be estimated by the difference in the results calculated from QHA and molecular dynamic (MD) simulations. By comparing the elastic moduli and density of MgSiO₃ bridgmanite obtained from QHA and MD within LDA, we also found only minor

differences ($< 0.5\%$) produced by the anharmonicity. Thus, these differences produced by different pseudopotentials and the anharmonic effect were adopted as the uncertainties of the calculated results ($< 0.8\%$). "

We also considered the effect of the MORB composition on elasticity. The uncertainties of calculated velocities and density of MORB estimated based on the errors for elastic properties ($< 0.8\%$) and the concentration of dilute substitutional solutes ($\pm 1 \text{ mol}\%$), are shown in the revised Figure 4.

Comment 3

Results of previous studies about the same topic, such as Tsuchiya (2011, PEPI) and Stixrude & Lithgow-Bertelloni (2012, Annu. Rev. Earth and Planet. Sci.), need to be assessed and compared carefully with the results of this study.

[Authors]: We added detailed discussion about previous studies on this topic. Please see the contents in lines 180-184 and 198-203:

"Tsuchiya (2011)²⁹ predicted negative V_P and V_S contrasts between MORB and pyrolite at the phase boundary of silica; however, the depth for such negative velocity perturbations is significantly shallower than our results, mainly because the temperature effect has not been taken into account²⁹."

"Stixrude and Lithgow-Bertelloni (2012)⁵¹ also found MORB has a faster V_S than the pyrolitic composition along the normal mantle geotherm at 1500-2500 km and the V_S contrast is comparable to our results; however, they did not find the shear softening of MORB at the mid mantle because they ignored the phase transition from stishovite to CaCl_2 -type silica."

Comment 4

This point is linked with the previous comment. Main part of this study is about the computation of seismic properties of the CF type phase. It is of importance, but the predicted behavior of this mineral turns out not to have controlling effects on the properties of basalt as a whole, unlike stishovite. Therefore, the authors need to clarify the most important contribution of this paper to the topic dealt with. They also need to

clarify technical barriers that have hampered accurate numerical evaluations of elastic properties of this mineral, and the innovative aspects of their approach to overcome the difficulties.

[Authors]: Experimental work on natural MORB suggested that MORB consists of approximately 30% Fe- and Al-bearing bridgmanite, 30% Ca-perovskite, 20% SiO₂, and 20% Fe-bearing CF-type phase below ~ 50 GPa. According to our results, MORB has distinctly slower wave velocities by up to ~ -8% for V_S and ~ -1.8% V_P at ~ 60 GPa (Fig. 4d and 4e), where stishovite transforms to the CaCl₂-type silica. Indeed, the elastic properties of SiO₂ have controlling effects on the velocity characteristics of MORB at ~ 60 GPa due to the softening of the shear modulus of stishovite at the phase boundary. However, the elastic properties of other phases including CF-type phases are indispensable data; otherwise, we cannot determine the velocities and density of MORB. After the phase transition of silica, the velocity characteristics of MORB are jointly controlled by the elastic properties of each phase. Thus, seismic properties of all phases are equally important for the determination of MORB's velocities and density because their volume proportions are close to each other. This work reports *ab initio* results for the elastic properties of calcium CF-type phases with substitutional solutes, which allows us to determine the velocity and density of oceanic crust by combining all mineral data at high P-T conditions together. Moreover, our work presents for the first time the depth-dependent velocity and density of natural oceanic crust at the conditions of the lower mantle. The significances/contributions of this study have been clearly clarified in the manuscript. In addition, we also added some contents to clarify technical barriers that have hampered accurate numerical evaluations of elastic properties of the lower-mantle minerals, and the innovative aspects of our approach used in this work to overcome the difficulties. Please refer to the content lines 279-287:

"The usual method used to calculate the elasticity at high temperature and pressure usually needs lots of vibrational density of states of material at different volumes and different strains (Wu and Wentzcovitch, 2011), which requires huge computational effort and hampers the accurate numerical evaluations of elastic properties at high P-T

conditions. Wu and Wentzcovitch (2011) developed a semi-analytical approach without requiring the vibrational density of states under strain by analyzing the relation between volume dependence of and strain dependence of the vibrational frequencies, which reduces the computational workload by two orders of magnitude compared to the usual method without loss of accuracy."

Comment 5

Minor points:

1. (Line 59): The scatterer mentioned is not beneath Tonga but beneath northern Peru (Haugland, et al, 2017).
2. (Figure 1): The symbols of experimental studies are rather difficult to see in Figure 1.
3. (Figures 4d and 4e): A horizontal line at $dV_p=0$ and $dV_s=0$ seems to help understand the points mentioned in the text.

[Authors]: Thanks for the suggestions. We correct the mistake in line 59, modified the symbols in Figure 1, and added horizontal lines at $dV_p=0$ and $dV_s=0$ in Figure 4.

Reviewer #2

Comment 1

This is an interesting study focusing on a very timely topic: the physical properties of subducted oceanic crust and their implications regarding compositional heterogeneity in the lower mantle. There is currently a strong debate about the lower mantle, which seems to be more complex and heterogeneous than previously thought. The mineral physics calculations performed in this study are much-needed for the interpretation of geophysical observations and models. However, I have some concerns as highlighted below.

[Authors]: Thanks for the recommendation. Please see our replies below.

Comment 2

I am concerned that none of the figures seem to show uncertainties, so it is difficult to

understand how well constrained the various properties are. The methods section should include some brief explanation of the uncertainties affecting the calculations and, more broadly, of the limitations of the methods employed. Clear statements on which properties are well/less well constrained should be included.

[Authors]: We estimated the uncertainties for the calculated elasticity and density of minerals by checking the quality of pseudopotentials used in this work and comparing the results of MgSiO₃ bridgmanite obtained from the quasi-harmonic approximation (QHA) and molecular dynamic (MD) simulations within LDA. Please refer to the contents in lines 295-308:

"To estimate the effect of pseudopotentials' quality on elastic properties, we also used harder pseudopotentials that includes semicore states with smaller core radii to conduct static calculations, which requires a cutoff energy of 400 Ry. We found that there are only minor differences (< 0.3%) in elastic moduli and density at static conditions when different pseudopotentials were used. Previous studies⁵⁹ on elastic properties and density of bridgmanite also reported minor differences in the results obtained using different pseudopotentials. The intrinsic anharmonicity ignored by QHA should be negligible at the lower-mantle pressure and temperature conditions and can be estimated by the difference in the results calculated from QHA and molecular dynamic (MD) simulations. By comparing the elastic moduli and density of MgSiO₃ bridgmanite obtained from QHA and MD within LDA, we also found only minor differences (< 0.5%) produced by the anharmonicity. Thus, these differences produced by different pseudopotentials and the anharmonic effect were adopted as the uncertainties of the calculated results (< 0.8%). "

We also considered the effect of the MORB composition on elasticity. The uncertainties of calculated velocities and density of MORB estimated based on the errors for elastic properties (< 0.8%) and the concentration of dilute substitutional solutes (± 1 mol%), are shown in the revised Figure 4.

Comment 3

Linked to the previous point, when reading the text comparing the new calculations

with experimental data (lines 107-111), the reader is left unclear about the reasons for discrepancies with the experimental data of Dubrovinsky et al., 2002 and why one should favor the measurements of Guignot and Andraut, 2004 (in practice it is really mostly these two studies that are being compared). Likewise, the reasons for discrepancies with experimental data at high pressure (Fig. S1) are not entirely clear. Again, this sort of brief discussion could be included in the Methods section along with a discussion of uncertainties and limitations, to give more confidence on the results presented and whether they really support the conclusions stated.

[Authors]: We have estimated the uncertainties for our results with the consideration of the limitations of the method used in this work. Please see the reply to the comment 2 above. Our calculated density of $\text{Na}_{0.4}\text{Mg}_{0.6}\text{Al}_{1.6}\text{Si}_{0.4}\text{O}_4$ CF-type phase agree well with experimental data in Imada et al. (2012) at low pressures but deviates from experimental measurements when pressure is higher than 80 GPa, above which the discrepancy between our results and Imada et al. (2012) increases with pressure. The deviation is probably due to the large non-hydrostatic pressure at high pressures since the high-pressure runs of Imada et al. (2012) were conducted at the lack of pressure medium. Particularly, Imada et al. (2012) also found apparent inconsistency among different experimental works, which was attributed the difference in the pressure scale used in the experiments. We added this reason in the caption of Figure S1:

"The calculated density of $\text{Na}_{0.4}\text{Mg}_{0.6}\text{Al}_{1.6}\text{Si}_{0.4}\text{O}_4$ CF-type phase agree well with experimental data in Imada et al. (2012) at low pressures but deviates from experimental measurements when pressure is higher than 80 GPa. This is probably due to the large non-hydrostatic pressure at high pressures since the high-pressure runs of Imada et al. (2012) were conducted at the lack of pressure medium."

In addition, the predicted pressure-dependent volumes of NaAlSiO_4 agree well with experimental measurements at 300 K in Guignot and Andraut (2004), but experimental data from Dubrovinsky et al. (2002) deviates from our results and other experimental data and the discrepancy is up to $\sim 2.5\%$. However, it is still unclear why there is a significant discrepancy among different experimental data and our results. Further researches are needed to clarify this problem and explaining the

discrepancy among experimental works is considerably beyond the scope of our calculations.

Comment 4

Lines 134-135: why is there a larger contrast in V_s than in V_p ?

[Authors]: MORB has distinctly slower wave velocities by up to $\sim -8\%$ for V_s and $\sim -1.8\%$ V_p at ~ 60 GPa (Fig. 4d and 4e), where stishovite transforms to the CaCl_2 -type silica. Such large negative velocity anomalies within MORB, which are mainly caused by the softening of the shear modulus (G) of stishovite at the phase boundary. Because this phase transition does not significantly affect the bulk modulus of silica, there is a larger contrast in V_s than V_p at ~ 60 GPa. Please refer to the statement in lines 175-177.

Comment 5

Fig. 4: of course, it is common practice to perform comparisons with PREM, but it could be useful to also carry out comparisons with 1-D profiles from various locations (e.g., near subducted slabs) from recent seismic tomographic models (e.g., Fukao and Kobayashi, 2013 for V_p ; Auer et al., 2014 and Chang et al., 2015 for V_s).

[Authors]: Thanks for the suggestion. Our results show that after the phase transition of silica, the MORB along the normal geotherm has relatively higher wave velocities than the ambient mantle (Fig. 3). The V_p and V_s perturbations are up to $\sim +1.5\%$ and $+2.0\%$, respectively. Meanwhile, seismic tomography (French and Romanowicz, 2015, 2014; Lekic et al., 2012) also found that the positive shear velocity anomalies in the mid-lower mantle beneath some local regions, which are generally regarded as the presence of subducted slab, can be larger than $+1.5\%$ and even to $+2\%$. When we compared our results with those seismic studies, we actually carried out comparisons with local tomographic models. Therefore, we cited more studies (Fukao and Kobayashi, 2013; Auer et al., 2014; Chang et al., 2015) that provide seismic tomographic models for the large positive velocity anomalies in the lower mantle to support our statements. Please refer to the contents in lines 50 and 203.

Comment 6

In order to make the text more accessible to a broad audience, it would be useful to add here and there the depth range that the results apply to (e.g., in line 30 of the abstract, and throughout the whole manuscript). For example, in which range of mid-mantle depths MORB may get lower velocities than ambient mantle? Which lateral variations may be expected?

[Authors]: We added the depth range where MORB has lower velocities than ambient mantle in the revised manuscript. In particular, MORB has distinctly slower wave velocities by up to $\sim -8\%$ for V_S and $\sim -1.8\%$ V_P at the depth where stishovite transforms to the CaCl_2 -type silica due to the softening of the shear modulus (G) of stishovite at the phase boundary. Consequently, the depth where large negative velocity anomalies occur is mainly controlled by the phase boundary between stishovite and the CaCl_2 -type silica, which strongly depends on temperature and alumina and water contents bearing in silica (Nomura et al., 2010). The incorporation of alumina plus hydrogen into silica can strikingly decrease the transition pressure, while increasing temperature significantly elevates it due to the positive Clapeyron slope (Nomura et al., 2010). Thus, the variations of "bright depth" for scattering (Kaneshima, 2016; Kaneshima and Helffrich, 2010, 2003), where the strong seismic scatterers were observed in the mid-lower mantle, may reflect the differences in temperature and Al_2O_3 and H_2O contents of silica in the oceanic crust. Please refer to the discussion in lines 184-193.

Comment 7

In order to make a stronger case on the importance and complexity of the lower mantle, the authors should also refer to recent studies of seismic anisotropy - e.g., in line 54 it could be added that there are not only plumes and slabs, but they can even complex interactions in the mid-mantle. For example, Chang et al., Nat. Comms., 2016 showed such example of complex interactions and it should be mentioned here.

[Authors]: We added the reference of recent anisotropic tomography in lines 54-55:

"Anisotropic tomography¹⁶ further suggested that there could be complex interactions between plumes and slabs in the mid mantle."

Comment 8

In order to make the study more appealing and easier to follow to a broad audience, it would be useful to include a final cartoon figure summarising observations of velocity heterogeneity that could potentially be explained in the framework of the calculations presented in this study.

[Authors]: We added a cartoon figure to summarize observations of velocity heterogeneities that could potentially be explained in the framework of our calculations. Please see the figure 5.

Comment 9

Overall the text needs to be improved; there are quite a few typos and the text should be sharper. For example, the sentence in lines 161-165 needs to be re-written for improved clarity, in line 99 write "ab initio CALCULATIONS", in line 141 write "with those" instead of "with these", in line 166 write "depend" instead of "depends", etc (these are just some examples, a thorough revision of the manuscript is needed).

[Authors]: Thanks for the suggestions. We carefully checked our manuscript for typos and editing problems and polished the English throughout the manuscript.

Comment 10

The last sentence of the main manuscript is quite vague - how could the calculations performed in this study help constrain, e.g., the vigour of mantle convection? More specific arguments would make the study stronger.

[Authors]: Our calculations demonstrate that subducted oceanic crust could not only produce a number of remnant fragments with several kilometers that are detected by seismic scattering in the lower mantle, but may also partially accumulate in the mid lower mantle or at the CMB to form the mesoscale chemical heterogeneities with positive velocity anomalies. How the subducted oceanic crust segmented from the

slab and accumulated or distributed in the lower mantle to further produce the seismic heterogeneities with different length scales needs the geodynamic modelling for the interaction between the subducted slab and the lower mantle, which is likely related to the vigor of mantle convection. However, how the mantle convection controls the length scales of subducted oceanic crust is still unclear. We added some contents in lines 249-251 and modified the last sentence in lines 253-256.

Reviewer #3

Comment 1

The paper reports on a study of the elastic properties of calcium ferrite-type phases with substitutional solutes, to determine the seismic velocities and densities of oceanic crust. The study is based on first principles methods, within the density functional theory implementation of quantum mechanics, and in particular by including the Hubbard U to address strong correlation. The paper contains an extensive set of calculations, and the results appear interesting, though I will not comment on their geophysical relevance. The first principles calculations have been validated on available experimental data, and so there is strong support that they are reliable. The authors are using tested methods, and so there would seem to be little doubt that these calculations are state of the art. However, very little detail of the molecular dynamics simulation is provided, and so it is difficult to make an informed judgement.

[Authors]: In this study, we predicted elastic properties of CF-type phases by using a semi-analytical approach proposed by Wu and Wentzcovitch (2011) rather than the molecular dynamic simulations. This method is based on the quasi-harmonic approximation (QHA) and has been successfully used to predict elastic properties of many minerals, including bridgmanite, ferropericlase, stishovite and CaCl₂-type silica, and corundum. Theoretical calculations within QHA ignore the anharmonic effects on elasticity and density. The anharmonic effect can be estimated by the difference in the results calculated from QHA and molecular dynamic simulations. By comparing the elastic moduli and density of MgSiO₃ bridgmanite from previous theoretical studies

within LDA, we found only minor differences ($< 0.5\%$) in the results obtained from the method used in this work and MD simulations. This indicates that anharmonic effects on elasticity and density at high P-T conditions are negligible. In addition, the anharmonic effect is also considered in the estimate of uncertainties for elasticity of minerals and MORB. Please refer to the contents in lines 295-308 and 331-334.

Comment 2

The manuscript should also be checked for editing problems, e.g.: line 258 "insufficiently describe"

[Authors]: We checked the manuscript and revised all typos and editing problems.

Reviewers' comments:

Reviewer #1 (Remarks to the Author):

Review on the revised manuscript of the paper "Velocity and density characteristics of subducted oceanic crust and the origin of seismic heterogeneity in the lower mantle" by Wang et al.

I think the authors adequately took my comments on the original manuscript of this paper into account. This paper can be accepted, with a few further revisions I suggest.

1. Lines 31-35 in Page 2.

"After this phase transition in silica, the oceanic crust has relatively higher wave velocities than those of the ambient mantle ..., indicating that the large shear velocity provinces (LLSVPs) unlikely originate from the subducted oceanic crust."

A very recent paper (Thomson, et al., 2019, nature, vol. 572, 643-647) states a proposition that is apparently contradictory to this statement, based on the measurement of shear and compressional wave velocities of Ca perovskite. I think that the authors should refer to this paper and discuss about the discrepancy.

2. Lines 75-78 in Page 4.

"we note that the estimated anomalies of the mid-mantle scatterers appear significantly larger than those expected for the oceanic crust (reference 29 better to be cited here?), probably because the elastic properties of relevant materials were calculated at static conditions and hence thermal effect cannot be taken into account."

Isn't it clearer to cite reference 29 (Tsuchiya, 2011) at the place I suggest, rather than at the end of this sentence?

3. Lines 186-188 in Page 10.

"however the depth for such negative velocity perturbations is significantly shallower than our results, mainly because the temperature effect has not been taken into account (reference 29.

Please discuss about possible amount of discrepancy between the results by Reference 29 (Tsuchiya, 2011) and those of the authors without temperature effects.

Reviewer #3 (Remarks to the Author):

The authors have satisfactorily answered all my scientific questions regarding the manuscript. However, the quality of the text can still be improved - for example, in some places there are "the" articles missing, whereas in others parts of the text they are not needed; in line 53 the text "are respective to the hot plumes" is odd, "scatterings" in line 61; in line 120 it should be "presented" instead of "present"; in line 156 it should be "of" instead of "on the"; in lines 166-167 "throughout the mantle" appears twice, etc etc -- there are many more examples in the rest of the manuscript, so the article needs a thorough revision by a native English speaker. Moreover, I could not find the new Figure 5 being mentioned in the main text.

Reviewer #4 (Remarks to the Author):

The authors have clarified the methodological point I was asking about. The paper is publishable in my opinion.

Reviewer #1

Comment 1

I think the authors adequately took my comments on the original manuscript of this paper into account. This paper can be accepted, with a few further revisions I suggest.

[Authors]: Thanks for the recommendation. Please see our replies below.

Comment 2

Lines 31-35 in Page 2.

“After this phase transition in silica, the oceanic crust has relatively higher wave velocities than those of the ambient mantle ..., indicating that the large shear velocity provinces (LLSVPs) unlikely originate from the subducted oceanic crust.”

A very recent paper (Thomson, et al., 2019, nature, vol. 572, 643-647) states a proposition that is apparently contradictory to this statement, based on the measurement of shear and compressional wave velocities of Ca perovskite. I think that the authors should refer to this paper and discuss about the discrepancy.

[Authors]: Thanks for the suggestion. We referred to this paper and another recent work on the velocities of Ca-perovskite (Gréaux et al. 2019, nature) and discussed the discrepancy in lines 241-255:

"A recent work conducted by Thomson et al. (2019)⁵³ suggested that subducted oceanic crust would be visible as low-seismic-velocity anomalies throughout the lower mantle when data is extrapolated to the lower-mantle conditions. The discrepancy between our results and Thomson et al. (2019)⁵³ resulted from the usage of different elastic and velocity data for Ca-perovskite. The calculated data used in this study is from previous *ab initio* molecular dynamic simulations³⁶, while the Ca-perovskite data adopted in Thomson et al. (2019)⁵³ was extrapolated from low pressure to the deep mantle conditions. Since velocities measured for Ca-perovskite^{53,54} are considerably lower than computational predictions at the conditions of the uppermost lower mantle, the extrapolated data would be expectedly lower than theoretical calculations³⁶ under deep mantle conditions. It is still unknown what results in the discrepancies in sound velocities of Ca-perovskite between

theoretical and experimental studies, and future research is needed to solve this problem. However, the uncertainties from extrapolation cannot be ignored because experimental measurements, especially high-temperature data, also show significant discrepancies^{53,54}."

Comment 3

Lines 75-78 in Page 4.

"we note that the estimated anomalies of the mid-mantle scatterers appear significantly larger than those expected for the oceanic crust (reference 29 better to be cited here?), probably because the elastic properties of relevant materials were calculated at static conditions and hence thermal effect cannot be taken into account." Isn't it clearer to cite reference 29 (Tsuchiya, 2011) at the place I suggest, rather than at the end of this sentence?

[Authors]: Thanks for the suggestion. We corrected the place for reference 29.

Comment 4

Lines 186-188 in Page 10

"however the depth for such negative velocity perturbations is significantly shallower than our results, mainly because the temperature effect has not been taken into account (reference 29).

Please discuss about possible amount of discrepancy between the results by Reference 29 (Tsuchiya, 2011) and those of the authors without temperature effects.

[Authors]: We also used the elastic data for all minerals at static conditions to estimate the V_p and V_s contrasts between MORB and pyrolite at the phase boundary of silica and found that our calculated results, both of negative velocity anomalies, are similar to those in Tsuchiya (2011). Please refer to the content lines 184-186:

"We obtained similar results (negative velocity anomalies and their depths) in Tsuchiya (2011)²⁹ when only elastic data at static conditions were used."

Reviewer #2

Comment 1

The authors have satisfactorily answered all my scientific questions regarding the manuscript. However, the quality of the text can still be improved - for example, in some places there are "the" articles missing, whereas in others parts of the text they are not needed; in line 53 the text "are respective to the hot plumes" is odd, "scatterings" in line 61; in line 120 it should be "presented" instead of "present"; in line 156 it should be "of" instead of "on the"; in lines 166-167 "throughout the mantle" appears twice, etc etc -- there are many more examples in the rest of the manuscript, so the article needs a thorough revision by a native English speaker. Moreover, I could not find the new Figure 5 being mentioned in the main text.

[Authors]: Thanks for the recommendation. We have made a professional reading service for English language editing of our manuscript. The new Figure 5 is also mentioned in lines 178, 223, and 241.

Reviewer #3

Comment 1

The authors have clarified the methodological point I was asking about. The paper is publishable in my opinion.

[Authors]: Thanks for the recommendation.